# Antibody-Loaded Nanoplatforms for Colorectal Cancer Diagnosis and Treatment: An Update

**DOI:** 10.3390/pharmaceutics15051514

**Published:** 2023-05-17

**Authors:** Rania Djermane, Celia Nieto, Milena A. Vega, Eva M. Martín del Valle

**Affiliations:** 1Chemical Engineering Department, University of Salamanca, Plaza de los Caídos s/n, 37008 Salamanca, Spain; r.djermane@usal.es (R.D.); celianieto@usal.es (C.N.); 2Biomedical Research Institute of Salamanca (IBSAL), University Care Complex of Salamanca, Paseo de San Vicente 58, 37007 Salamanca, Spain

**Keywords:** colorectal cancer, nanomedicine, monoclonal antibodies, biosensors, nanoimaging, drug delivery systems, combination nanotherapy, theranostics

## Abstract

At present, colorectal cancer (CRC) is the second deadliest type of cancer, partly because a high percentage of cases are diagnosed at advanced stages when tumors have already metastasized. Thus, there is an urgent need to develop novel diagnostic systems that allow early detection as well as new therapeutic systems that are more specific than those currently available. In this context, nanotechnology plays a very important role in the development of targeted platforms. In recent decades, many types of nanomaterials with advantageous properties have been used for nano-oncology applications and have been loaded with different types of targeted agents, capable of recognizing tumor cells or biomarkers. Indeed, among the different types of targeted agents, the most widely used are monoclonal antibodies, as the administration of many of them is already approved by the main drug regulatory agencies for the treatment of several types of cancer, including CRC. In this way, this review comprehensively discusses the main drawbacks of the conventional screening technologies and treatment for CRC, and it presents recent advances in the application of antibody-loaded nanoplatforms for CRC detection, therapy or theranostics applications.

## 1. Introduction

Despite our knowledge of colorectal cancer (CRC) having improved considerably, it is still the second leading cause of death from cancer globally. CRC was responsible for almost 1 million deaths (55.1% males and 44.87% females) worldwide in 2020, and it is expected that the mortality rate from this type of cancer will continue to increase in the coming decades due to elevated exposure to environmental risk factors [1].

The high CRC mortality rate, which is only surpassed by that of lung cancer, is partially caused by the distant metastatic nature of colorectal tumors [1,2]. Since CRC is a silent disease and colon tumor cells have a great invasive capacity, people usually do not develop clinical manifestations until tumors are metastasized. Consequently, nearly a quarter of CRCs are detected in advanced stages, although the implementation of screening programs in many countries has made it possible to improve their early diagnosis [3,4].

In relation to the latter, CRC is classified into various stages, which are summarized in Figure 1. Briefly, in stage 0, malignant cells arise from the colonic wall mucosa and start to proliferate. Stage I begins when the malignant tumors, in addition to being in the wall mucosa, also emerge in the submucosa and the muscularis propria. Stage II occurs when cancer cells progress further into the pericolorectal tissues, visceral peritoneum and attached organs. Then, when cancer cells metastasize to the nearby lymph nodes and tissues, stage III of CRC starts. Finally, stage IV begins when metastases appear in other organs, such as the ovaries, liver or lungs [5,6].

If colorectal tumors are detected when they are localized (stage I), the chance of patients surviving at 5 years is over 90%. However, if they are diagnosed at stage II, the survival rate drops to 50%, and if tumors have metastasized to other organs by the time they are detected (stage IV), the probability of survival is only 10% [3,5,6,7]. Consequently, there is an urgent need to develop novel strategies that allow the early diagnosis of this complex disease, as well as to develop more effective therapies for metastatic CRC (mCRC), which has the most fatal prognosis.

When mCRC lesions appear, often in the liver and lungs [8], curative surgery becomes complicated, and patients must receive radiotherapy and chemotherapy in an attempt to achieve tumor shrinkage and suppress further tumor spread and growth. Generally, chemotherapy is the backbone of mCRC treatment, and the most commonly used chemotherapeutic agents to combat this type of cancer, as will be discussed later, are 5-fluoracil (5-FU), oxaliplatin, irinotecan (CPT-11), celecoxib and capecitabine. These drugs can be administered alone or in combination to increase their efficacy. Nonetheless, both single and combined therapies have severe dose-related systemic toxicities. In addition, chemotherapy has other major, well-known drawbacks, such as low tumor specificity, the emergence of unpredictable resistance and, consistently, poor response rates [4,9,10].

For this reason, great efforts have been made over the last two decades to develop novel, targeted therapeutic agents capable of directly inhibiting the proliferation, differentiation and migration of CRC cells. These three processes are mainly mediated by the signaling pathways shown in Figure 2 and, because of this fact, most of the developed targeted therapies inhibit the molecules that are involved in them [4,11,12,13].

As will be discussed later in this paper, at present, the majority of the targeted therapeutic agents approved by the U.S. Food and Drug Administration (FDA) and the European Medicines Agency (EMA) for the treatment of CRC are monoclonal antibodies (mAbs) [4,14]. The clinical administration of these biological drugs, which can recognize specific cell surface receptors or membrane-bound sites and regulate downstream molecular pathways [4], has improved the prognosis for many patients. However, as in the case of traditional chemotherapeutics, these targeted agents must be intravenously administered. Otherwise, if they were administered orally, the pH variations in the gastrointestinal tract would impede their pharmacological activity. As a result, since drugs administered by the parenteral route must reach systemic circulation before arriving at their site of action, mAbs also have marked side toxicity, which limits the dose of the biological drugs that patients can receive [14,15].

Due to the above-mentioned issues, there is still a constant demand for improved CRC diagnostic approaches and therapies, and, in this sense, nanomedicines have emerged as a promising tool [14,16,17]. The development of nanotechnology has been considerable in recent years, and this has allowed the design of novel diagnostic, therapeutic and theranostic nanosystems for the successful management of different types of cancer, including CRC [9,17,18]. In this way, nanomaterials are being used to develop contrast agents for biomedical imaging applications and to detect specific colon tumor biomarkers. Additionally, nanomaterials are being used to improve the efficacy of colon cancer thermotherapy and phototherapy. In the same manner, smart nanocarriers, which are capable of targeting surface molecules that are specific to malignant colon cells or releasing their payload only under certain microenvironment conditions, are being developed as novel drug delivery systems (DDSs) to reduce the systemic toxicity that characterizes conventional antitumor drugs (Figure 1) [16].

Therefore, this article summarizes advances in the development of nano-approaches for the diagnosis and therapy of CRC that are based on the use of the main targeted ligands approved by the drug regulatory agencies: mAbs. Initially, after reviewing the strategies currently applied to diagnose and treat CRC, the main types of nanoplatforms used for the same purpose are briefly summarized. Information on how mAb-loaded nanosystems are capable of targeting tumor cells is also provided. Finally, an overview of the mAb-loaded nanoplatforms developed in recent years for CRC diagnosis, targeted therapy and theranostic applications is discussed.

## 2. Current Strategies for Colorectal Cancer Diagnosis and Therapy

### 2.1. Current Methods for CRC Screening and Detection

As mentioned in the previous section, in recent years, many countries have implemented screening programs to detect colon carcinomas in their early stages in asymptomatic individuals. Within these screenings, the most frequently used techniques are stool tests, colonoscopies, computed tomography (CT) colonography, sigmoidoscopies and double-contrast barium enemas.

As far as stool tests are concerned, they are usually based on kits that allow the detection of very small amounts of blood in the stool. These tests are faster and less expensive compared to other screening techniques, but they do not detect non-bleeding adenomas or polyps. In contrast, colonoscopies can detect small, non-bleeding polyps, during which the entire colon and rectum can be screened using a flexible tube light with a lens, which also has a tool that allows excising abnormal tissues. However, this diagnostic technique requires prior cleansing of the colon and sedation, and it may cause serious bleeding or tears in the intestinal wall. As the latter is an inconvenience for many patients who cannot receive anesthesia, CT colonography, sigmoidoscopies and double-contrast barium enemas are used as an alternative, although they are not as accurate. Colonography, which uses a CT scanner to take images of the colon, is not invasive but sometimes does not allow for clear differentiation between polyps and other abnormalities. On the other hand, although sigmoidoscopies have a great degree of accuracy, they are not useful for detecting polyps that are outside the end of the colon or the rectum. Likewise, barium enemas, which allow X-ray imaging, may generate false positive results and also require colon cleansing [6,19,20].

Thus, as it can be seen, even though all diagnostic techniques are very useful, and that colonoscopy especially aids in reducing CRC mortality [19], all methods have remarkable drawbacks, which are summarized in Table 1. Thus, given the importance of the early diagnosis of malignant colon tumors for patients’ overall survival (OS), the development of new, sensitive, faster, non-invasive and low-cost alternatives for CRC detection is clearly needed. For this reason, as will be explained later, more and more diagnostic alternatives based on biomarkers, as well as new imaging approaches, have emerged recently [19,21].

### 2.2. Current Options for CRC Therapy

As already mentioned in the Introduction, there are three alternatives for CRC treatment at present, which are applied in isolation or in combination depending on the stage in which tumors are found. The first is surgical resection, which is an effective option for localized, small tumors, i.e., stage I tumors. Otherwise, the second alternative involves radiotherapy, in which X-rays are used to eliminate cancer cells when surgery is not enough. Nonetheless, despite being effective, radiotherapy has important adverse effects, such as skin irritation, nausea, tiredness, urinary incontinence and tissue adhesions. Finally, the third alternative, chemotherapy, is the only possible option for patients with tumors in more advanced stages [9,20].

To date, the FDA has certified around sixteen antitumor compounds for CRC treatment, of which five are conventional chemotherapeutic drugs that generally act by arresting the cell cycle. As stated in the Introduction, these drugs are 5-FU, CPT-11, oxaliplatin, celecoxib and capecitabine. Capecitabine and 5-FU inhibit the enzyme that catalyzes the production of thymidine, which is one of the DNA nucleotides; oxaliplatin disrupts the replication and transcription processes; CPT-11 blocks the action of the topoisomerase I, which is essential for DNA replication; and celecoxib is capable of inhibiting COX-2, which in turn is responsible for the synthesis of the vasodilator PGI2 (Table 2) [20,22]. Since some phase II–III studies in patients with mCRC showed clear significant survival advantages for some of these agents when combined with others, all the drugs can be administered in combination to achieve a greater antitumor efficacy, as in the FOLFIRI, FOLFOX, CAPOX and XELOX regimens. In the first regimen, CPT-11 is combined with 5-FU and leucovorin (LV), which boosts the antitumor activity of 5-FU by inhibiting thymidylate synthase. LV and 5-FU are also combined in the FOLFOX regimen, in which oxaliplatin is also administered. Finally, in the CAPOX and XELOX regimens, oxaliplatin and capecitabine are co-administered [23,24,25,26].

Regarding the administration of these compounds, either in isolation or in combination, it should be noted that it must be carried out parentally most of the time. This, along with the fact that traditional chemotherapeutics also kill normal cells that proliferate faster due to their lack of specificity, causes important side effects. The most relevant ones are summarized in Table 2 [20,27].

Hence, in order to specifically inhibit the growth and spread of colon tumors and reduce chemotherapeutic side toxicities, targeted agents that are able to interfere with concrete molecules presented on the CRC cell surface or are involved in their proliferation and dissemination have been investigated in depth in recent decades [20,28].

The first targeted compound to be approved by the FDA for mCRC treatment in combination with 5-FU was bevacizumab (Avastin^®^), a mAb capable of recognizing the vascular endothelial growth factor (VEGF) [29]. This factor, when bound to its receptors (VEFGR-1 and VEFGR-2), promotes the proliferation of endothelial cells and the formation of blood vessels, thus reducing tumor interstitial pressure [30]. For this reason, in addition to bevacizumab, other compounds targeting VEGF or its receptors were developed later. In fact, some have already been approved by the main regulatory agencies for administration in combination with traditional chemotherapeutic drugs for CRC therapy, such as the recombinant fusion protein aflibercept, the mAb ramucirumab and the multi-targeting kinase inhibitor regorafenib (Figure 3A) [9,31,32]. Moreover, some other targeted agents that have been approved for use in the clinical setting are cetuximab, panitumab, nivolumab, ipilimumab and pembrolizumab [9]. All are mAbs, but while the first two recognize the epidermal growth factor receptor (EGFR), nivolumab and pembrolizumab bind to the programmed death 1 (PD-1) molecule, and ipilimumab binds to the cytotoxic T-lymphocyte antigen 4 (CTLA-4) [30,33,34,35,36]. On the one hand, EGFR is overexpressed in 25–77% of CRCs, and as shown in Figure 2 and Figure 3B, it is related to a higher rate of tumor cell proliferation [4]. On the other hand, PD-1 and CTLA-4 down-regulate the effect of T-cell activity and enable cancer cells to escape for antitumor immunosurveillance [37,38]. Accordingly, administering mAbs capable of inhibiting these three molecules together with cell cycle inhibitors has been shown to increase the survival rate of patients with CRC [20].

Targeted drugs currently approved by the FDA and EMA for CRC treatment are listed in Table 3, and as can be noticed, most of them are mAbs. Of the different targeted drugs that are available at present, mAbs have been more extensively characterized in comparison to other molecules, such as peptides and aptamers [39], and they are the most used targeted drugs in the clinical setting. For this reason, to date, these immunoglobulins are the agents that are more frequently used to functionalize nanomaterials for CRC diagnosis and treatment [40]. Thus, from this section onward, this review focuses on these mAb-loaded nanosystems.

## 3. Most Researched Nanosystems for the Diagnosis and Treatment of CRC

Although new biocompatible and biodegradable materials are constantly being developed for cancer diagnosis and therapy, in general, all designed nanosystems can be classified into two types: organic and inorganic nanoplatforms. Among them, the nanosystems that use the most mAbs for CRC applications are summarized in Figure 4 [41].

### 3.1. The Most Investigated Inorganic Nanoplatforms for CRC Applications

At present, a variety of inorganic materials have been used to fabricate nanosystems for medical applications, with the most common nanoplatforms being gold, silver, magnetic, and silica nanoparticles, quantum dots and carbon nanotubes [41,42].

Inorganic nanosystems have unique magnetic, electrical and optical properties that make them well-suited for diagnostic and theranostic applications. In addition, their geometry can vary greatly, they have a large surface area and, normally, their surface can be easily functionalized with targeting ligands, so they are widely used to develop new drug delivery systems (DDSs) [41,43]. For instance, gold nanoparticles (AuNPs), which are simple to produce with great control of size and shape and are biocompatible, have a strong near-infrared (NIR) light absorption capacity and efficient light-to-heat conversion. Additionally, AuNPs allow secondary modifications on their surface [41,44]. In a similar way, magnetic nanoparticles (MNPs) are excellent candidates for hyperthermia applications and magnetic resonance imaging (MRI) and, moreover, they can be used to specifically deliver drugs to tumor tissues due to their magnetic field-responsive properties [43,45]. Additionally, mesoporous nanoparticles (NPs) of silica (MSNs) have a controlled pore size and morphology and large surface area, and since they can be easily functionalized, their pores are often used to carry drugs [43,46]. Similarly, carbon nanotubes (CNTs), which are constructed in three-dimensional sp^2^ hybridizations, have attracted researchers seeking to develop new DDSs because of their extremely high drug-loading efficiency and structural flexibility [47]. Finally, quantum dots (QDs), which are cheap to produce and can also be functionalized, can emit photons of several wavelengths upon excitation, depending on their size and shape. Thus, they are widely used for targeted diagnostic imaging of gastrointestinal (GI) cancer [42].

### 3.2. The Most Investigated Organic Nanoplatforms for CRC Applications

As with inorganic materials, a large number of organic materials have also been investigated to produce novel nanoscale systems for medical applications [48], which normally have fewer biodegradability and biocompatibility issues than inorganic materials [41,49]. In general, organic nanoplatforms can be classified into two subtypes: polymeric or lipid in nature [41].

Regarding the former, it should be noted that polymeric nanosystems are colloidal dispersions of either natural, synthetic or pseudo-synthetic origin that can incorporate different compounds by surface adsorption, intra-NP encapsulation or surface conjugation [49,50]. The most common techniques used to prepare polymeric nanoparticles are emulsification methods (single-emulsion or double-emulsion solvent evaporation techniques), although nanoprecipitation, ionic gelation and microfluid techniques are also widely used [41]. At present, some of the most used polymers are gelatin, alginate, chitosan, collagen, hyaluronic acid (HA), polyethylene glycol (PEG), polylactic acid (PLA), polyglycolic acid (PGA), poly-lactic-co-glycolic acid (PLGA) and poly (*ε*-caprolactone) (PCL) [49]. All of them allow the production of different polymeric structures, such as capsules (polymer–drug conjugates), amphiphilic nanoparticles (micelles), hyperbranched molecules (dendrimers) or polymer networks (nanohydrogels) (Figure 4), which are diverse in terms of the compounds that can be loaded into them [41,51]. In this way, since polymeric nanoparticles can carry compounds in their core and are homogeneously distributed in their matrix, chemically conjugated or attached to their surface, they can be used to transport both hydrophilic and lipophilic molecules with small to large molecular weights [41].

Secondly, lipid-based nanoplatforms are generally composed of layers of safe lipolipids [49,52] that typically generate spherical-shape structures integrated by, at least, one internal aqueous compartment surrounded by one or more lipid bilayers. Nonetheless, several structures of lipid-based nanoplatforms can be found, such as liposomes and solid lipid NPs (Figure 4) [41]. On the one hand, liposomes are spherical vesicles synthesized with one or more layers of amphiphilic phospholipids and cholesterol, which provides stability to the bilayer structure. In this way, these vesicles permit the encapsulation of bioactive compounds of a different nature within their empty core or their layers [49,53]. Solid lipid NPs (SLNs) are colloidal nanoparticles prepared with fatty acids, fatty alcohols and different glycerides [53]. Thanks to their solid matrix, they possess improved stability, and they are good carriers of both hydrophilic and lipophilic bioactive compounds, such as liposomes [52,54]. Moreover, their production is easier to scale up compared to that of the other nanoplatforms [41,52,54].

## 4. Nanoplatform Pathways for Targeting Tumor Tissues

Before providing specific examples, it should be mentioned that one of the advantages of using mAb-loaded nanosystems for CRC diagnosis and therapy is that these nanoplatforms can target tumor tissues and protect healthy cells from harmful substances using two different approaches: passive and active targeting [41,55] (Figure 5).

### 4.1. Passive Targeting of Tumor Tissues

In the late 1980s, Maeda and co-workers observed that, due to tumor angiogenesis, in which the rapid development of new and irregular blood vessels occurs, tumor vessels display a discontinuous epithelium that allows macromolecules larger than 30 kDa to be preferentially distributed to the tumor interstice [56,57,58]. In addition, these authors realized that, while healthy tissues keep the extracellular fluid constantly drained to lymphatic vessels, this lymphatic function is defective in tumors, causing macromolecules and nanoparticles with diameters of up to 600 nm to accumulate after not being efficiently cleared [41,59] (Figure 5A). A few years later, this observation was termed the enhanced permeability and retention (EPR) effect and, since then, it has become the vehicle for many scientists to deliver nanoplatforms to malignant tissues for medical purposes [57,60].

However, despite the above-mentioned advancements, the effect of EPR is sometimes limited because it depends on tumor vasculature, which can vary depending on the tumor type and the anatomical site. For instance, prostate and pancreatic tumors are hypovascularized, and the high tumor interstitial fluid pressure (IFP) that characterizes malignant tissues often hinders the homogeneous distribution of nanosystems [59,61]. For this reason, the optimal way to target cancer tissues is using a combination of both passive and active targeting approaches.

### 4.2. Active Targeting of Tumor Tissues

Active targeting relies on the modification of nanosystem surfaces using targeting ligands with affiliation to receptors only expressed/overexpressed by tumor cells. Therefore, this approach allows the delivery of a certain amount of diagnostic or therapeutic agents to the targeted disease area (Figure 5B). The targeted ligands, in addition to mAbs, include aptamers, proteins, enzymes and polysaccharides [41,55,62]. Nonetheless, functionalizing nanosystems with immunoglobulins has the advantage that the administration of many of them in the clinical setting has already been approved. Moreover, in addition to serving as target agents, mAbs have cytotoxic effects relevant to therapeutic applications [61]. The cytotoxicity of mAb can be mediated by four mechanisms: (i) complement-dependent cytotoxicity (CDC), (ii) antibody-dependent cell-mediated cytotoxicity (ADCC), (iii) antibody-dependent cellular phagocytosis (ADCP) and (iv) complement-dependent cellular cytotoxicity (CDCC) [61,63,64,65].

## 5. Antibody-Loaded Nanosystems for CRC Diagnosis

As already noted, diagnosing colon cancer before it metastasizes to nearby lymph nodes or other tissues is essential to provide the best chance for successful cancer therapy [41]. In order to overcome the challenges in current screening methods, various mAb-loaded nanosystems have been produced and evaluated for their efficacy in three main CRC diagnostic applications: (i) detection of extracellular tumor biomarkers, (ii) detection of circulating tumor biomarkers and (iii) cancer imaging (Figure 6) [41,42,66].

### 5.1. Antibody-Loaded Nanosystems for the Detection of Extracellular CRC Biomarkers

Tumor biomarkers can be defined as biological molecules (proteins, protein fragments, nucleic acids, carbohydrates or antibodies) produced by an organism in response to the existence of a tumor. They can be found in the blood, saliva, urine, stool or malignant tissues, and their determination can be useful for early cancer detection. Nevertheless, biomarker levels are usually very low in the early stages of tumor progression, so developing more sensitive detection methods is required [41]. Thus, producing new sensors for cancer biomarkers is one of the many objectives currently pursued in the nanotechnology field, and mAb-loaded nanosystems can be a very advantageous option to achieve this aim.

In the specific case of CRC, there is an extracellular biomarker qualified by the FDA: the carcinoembryonic antigen (CEA) [41,67]. It is an oncofetal glycoprotein expressed in mucosal cells but overexpressed in several types of carcinomas, including colorectal carcinomas. Its quantification in the blood serum can be helpful for CRC diagnosis and can serve as an indication of the cancer development stage, but current analytic approaches (fluorescence immunoassays, electrochemistry and enzyme-like immunoabsorbent assays (ELISA)) for CEA detection have several disadvantages. For example, most of these approaches require expensive reagents and are based on cumbersome experimental protocols that take a long processing time [68,69].

To address these issues, Zhang et al. developed a new electrochemiluminescence (ECL) immunosensor for CEA detection based on the use of flower-like gold nanoparticles coated with a thin layer of bovine serum albumin (BSA) (Au@BSA NPs). They also incorporated luminol and an anti-CEA antibody into their coated NPs. In this way, when their immunosensor captured CEA antigens, the formed antibody–antigen immunocomplex hindered the interfacial electron transfer and influenced the ECL reaction between luminol and H_2_O_2_, which was used as a co-reactant. Therefore, the ECL intensity diminished with the increase in the CEA concentration. The developed immunosensor had a high sensitivity (with a 0.0003 ng/mL detection limit), wide linear range and great stability and reproducibility, so it may be a fast and simple alternative for CEA detection in the future [68].

With a similar goal, Zheng et al. developed a new photoelectrochemical CEA biosensor using porous hollow NiS@NiO/TiO_2_ spheres with outstanding photoelectric conversion efficiency. Once developed, these authors used these spheres to modify an electrode surface, which received BSA and an anti-CEA antibody. They successfully obtained a stable CEA biosensor with a wide linear detection range (0.001–45 ng/mL), showing the excellent potential of hollow NiS@NiO/TiO_2_ spheres in clinical diagnostic applications [70].

In another work by Jozghorbani and co-workers, a new, label-free electrochemical immunosensor for CEA detection was developed from graphene oxide. After simply coating a glass carbon electrode with partially reduced graphene oxide and activating its carboxyl groups using EDC/NHS chemistry, Jozghorbani et al. loaded anti-CEA antibodies on the electrode surface without any steric hindrance. BSA was used to block non-specific sites on the modified surface and, as a result, the authors obtained an immunosensor that provided results similar to those obtained with the ELISA technique but without the need to use expensive and sophisticated instruments for CEA determination [71].

### 5.2. Antibody-Loaded Nanosystems for the Detection of Circulating CRC Cells and Tumor DNA and microRNA

Since almost 30% of colorectal carcinomas do not produce CEA [72], an increasing amount of research has been performed in recent years on new types of circulating tumor biomarkers, such as circulating tumor cells (CTCs), tumor DNA (ctDNA) and microRNA (miRNA) [73]. On the one hand, CTCs are epithelial cancer cells with distinct nuclear morphology that enter the circulatory system and can be detected in the peripheral blood [72,74]. Since these cells can function as an early indicator of potential metastasis, the ability to detect CTCs before the development of secondary tumors may help to improve patient outcomes [41]. On the other hand, the blood contains minute amounts of ctDNA derived from primary and secondary lesions that may provide an overview of the mutations that a patient’s tumor can harbor, thus leading to the diagnosis of CRC in asymptomatic individuals [74]. In a similar manner, miRNAs can also be found in the circulatory system, either alone or in combination with some proteins. These single-stranded RNA chains regulate the expression of hundreds of gene targets post-transcriptionally and, since they are involved in many biological processes, such as cellular proliferation, differentiation and apoptosis [74], they can provide relevant information about tumor growth [41]. In this manner, circulating biomarkers can provide useful, real-time information on the condition of patients. However, since they exist in low concentrations and have high heterogeneity, new sensing methodologies with great sensitivity, multiplex detection and tolerance to complex backgrounds are needed to promote the clinical application of circulating biomarkers [73].

An example of the latter is the FDA-approved commercial CellSearch^®^ kit for the enumeration of CTCs in patients with mCRC, which is based on the use of a mAb against the epithelial cell adhesion molecule (EpCAM), a surface protein expressed by most carcinoma epithelial cells. This CTC detection kit is already in clinical use, but it has some limitations since EpCAM expression is absent in some cancer cells, such as those that undergo epithelial-to-mesenchymal transition. The kit requires large volumes of blood and is quite expensive, so CellSearch^®^ is not applied routinely [72,74,75,76]. To address these drawbacks, various technologies were developed, such as AdnaTest^®^, another commercialized positive selection platform. It relies on the use of immunomagnetic beads coated with a cocktail of mAbs for the enhanced capture of CTCs, which are later validated using multiplex real-time polymerase chain reaction (PCR) for various gene panels. In this way, using the combination of two methodological approaches, this technique has improved sensitivity and can characterize more heterogeneous CTC samples than CellSearch^®^ [76].

Additionally, regarding ctDNA detection, it should be noted that quantification of the ctDNA methylation level can effectively determine the degree of tumor malignancy. Nonetheless, the existing methods for analyzing it (such as PCR, sequencing and microarrays) require pretreatment of ctDNA. In contrast, anti-5-methycytosinine (5-mC) antibodies can be directly immobilized on electrodes using a covalent coupling method to capture methylated ctDNA without sample pretreatment [77]. This alternative was carried out by Povedano et al. in 2018, who developed two different electrochemical affinity biosensing approaches for 5-mC in DNA that do not require bisulfite or PCR testing [78]. Likewise, more recently, Hanoglu et al. reported an electrochemical sensing system based on MNPs that enabled quantification of the methylated septin-9 gene, which is considered a biomarker in early CRC [79].

Finally, regarding the detection of circulating miRNA for CRC diagnosis, it is worth mentioning that most biosensors have been designed to quantify the members of the miR-17-92-miRNA cluster, one of the best-known oncogenic miRNA clusters involved in colorectal carcinogenesis [74]. However, the majority of these biosensors are not based on the use of mAbs [80]. Immunoglobulins are more frequently used to develop immunosensors capable of detecting exosomes, lipid-membrane vesicles that provide intercellular communication involving proteins, lipids, DNA, mRNA and miRNA. Since these vesicles also derive from tumor cells, they are considered as other biomarkers [80], for which several immunosensors can be already found in the literature. For instance, Ortega and co-workers recently designed an electrochemical microfluidic immunosensor for claudin-7 biomarker determination in circulating exosomes. They immobilized anti-claudin-7 mAbs on NPs with a metal–organic framework, which were covalently anchored to the central channel of a microfluidic immunosensor. Then, they used a horseradish peroxidase (HRP)-conjugated secondary antibody to generate a current whose magnitude was directly proportional to claudin-7 sample levels in a linear range, from 0.002 to 1 ng/mL [81].

### 5.3. Antibody-Loaded Nanosystems for CRC Imaging

In addition to the development of biosensors, new antibody-loaded nanosystems have also been produced to improve current imaging techniques for colorectal tumors. Of the different nanoplatforms mentioned in Section 3, the most used for this application are MNPs, QDs and lipid-based NPs [42].

Several years ago, Gazouli et al. assessed the CRC-targeting ability of bevacizumab-conjugated QDs in vivo, finding that the developed nanosystem was effectively useful to detect VEGF-expressing tumors using fluorescence imaging. Thus, these authors opened new perspectives for VEGF-targeted noninvasive imaging in the clinical setting [82].

In a similar manner, Cho et al. reported the synthesis of iron oxide NPs, coated with silica and loaded with cetuximab and an organic dye (rhodamine B isothiocyanate), for MRI imaging of CRC cells overexpressing EGFR. In their work, they were able to amplify the local concentration of the NPs using an external magnetic field and, as a result, they could image non-invasive colon cancer xerographs in mice [83].

Portnoy et al. also incorporated cetuximab into a nanoplatform for NIR imaging of CRC. These authors developed liposomes and attached the mentioned mAb and fluorescent molecule indocyanine green using passive adsorption. After performing in vitro experiments, Portnoy and co-workers found that the synthesized liposomes had a larger quantum yield than the free indocyanine green and specific tumor cell recognizing ability, so they suggested the in vivo evaluation of their nanosystem for future diagnostic purposes [84].

## 6. Antibody-Loaded Nanosystems for the Therapy of CRC

In recent years, surface-engineered NPs have attracted fast-growing interest due to their ability to potentially overcome some of the most important limitations in current antitumor therapies, such as non-specific biodistribution, inefficient accumulation in tumor tissues and the apparition of multidrug resistance (MDR) [18,41,85]. Regarding the specific matter of CRC, more than one hundred studies have been published concerning the preclinical evaluation of nanomedicines [18]. The most common therapeutic applications of nanomedicines conjugated with mAbs are the ones detailed below and summarized in Figure 7: (i) the development of novel drug DDS and (ii) the development of combination therapies.

### 6.1. Targeted DDS for the Treatment of CRC

Undoubtedly, one of the most important applications of mAb-loaded nanosystems is the production of novel targeted DDS since they can guide the chemotherapeutic agents to the tumor site while improving their pharmacokinetics and protecting them from degradation and clearance [41,85,86]. Thus, in recent years, numerous research papers have been published in which some of the mAbs listed in Table 3 have been incorporated into nanosystems with different natures to develop targeted nanocarriers for CRC therapy (Figure 7A).

For instance, in 2021, Feng and co-workers developed carboxymethyl chitosan and PLGA NPs as carriers of docetaxel and methoxy-PEG-poly(*β*-amino ester) NPs as bevacizumab transporters to improve the intestinal absorption of both drugs when administered orally. They physically mixed the two NPs, which had a pH-sensitive drug release properties, and they verified that the achieved binary mixture of NPs increased the bioavailability of docetaxel and bevacizumab and that it had excellent cytotoxic activity on colon cancer cells [87].

Akborzadeh-Khiavi et al. synthesized PEGylated AuNPs conjugated with cetuximab to induce ROS-dependent apoptosis in CRC cells with a mutation in *Kras*, a major oncogene driver in CRC progression. The engineered nanoconjugates were capable of interfering with redox homeostasis in tumor cells, resulting in a detrimental accumulation of ROS and consequent tumor growth inhibition. In this way, the developed nanosystem may be considered for the effective treatment of not only CRC but also other solid tumors [88].

Likewise, Salim et al. reported the preparation of egg serum albumin NPs loaded with cetuximab. They synthesized the NPs using a simple improved desolvation method and used glutaraldehyde as a crosslinker agent to load the mAb. Compared to the administration of free cetuximab, the NPs prepared by these authors had improved apoptotic and antiproliferative actions, so they could be suggested for the targeted administration of antitumor medicines in the near future [89].

Finally, Marcelo et al. also incorporated cetuximab into a nanocarrier, but their method was based on the use of MSNs. In addition to mAb, they also modified the NPs with a second targeted agent (an acetazolamide derivative), and the resulting nanosystem was used to transport a combination of two broad-spectrum chemodrugs: doxorubicin and ofloxacin. The results obtained by these authors showed that the developed nanocarrier had a great cellular uptake rate and significant toxicity to the HTC116 colon cancer cell line, so it also has a high potential to be used as a novel nanotherapy agent [90].

### 6.2. Antibody-Loaded Nanosystems for CRC Combination Therapy

Given that tumor cells often acquire resistance against chemotherapeutic drugs, multimodal treatment approaches combining more than one therapeutic modality within a single nanoplatform (Figure 7B) have received considerable attention in recent years. They have also been favored by the fact that multimodal nanosystems have proven to be more effective for better overall therapeutic responses [91].

For the specific case of CRC, Fang et al. recently reported the synthesis of cetuximab-conjugated, hybrid lipid–polymer NPs with photothermal properties as NIR-triggered nanocarriers for irinotecan. The investigations performed by the authors demonstrated that this mAb-functionalized dual therapeutic nanosystem had good tumor-targeting ability and enhanced anticancer therapeutic activity compared to therapeutic strategies (photothermal therapy and chemotherapy) applied in isolation, so it may represent a promising multifunctional nanoplatform for CRC therapy [92].

Similarly, Conde and his colleagues developed a hydrogel patch for the local treatment of CRC using a triple-combination therapy: gene, drug and photothermal therapy. They synthesized a tunable hydrogel patch impregnated with bevacizumab-loaded gold nanorods, which had the capacity to convert NIR radiation into heat, and with gold nanospheres carrying siRNA molecules that silence *Kras*. Using in vivo assays, Conde et al. demonstrated that the hydrogels, developed with a prophylactic aim, could cause tumor remission when applied to non-resected tumors and avoid tumor recurrence when applied following tumor resection, so they proposed modifying this local, triple-combination therapy to treat other cancer cell types [93].

## 7. Antibody-Loaded Nanosystems for CRC Theranostic Applications

Although new nanosystems for the diagnosis of CRC (and other types of tumors) continue to be developed, there has been an increasing tendency in recent years to combine diagnostic and therapeutic features in the same platform, i.e., in theranostic devices. In this manner, targeted imaging, once integrated into these theranostic systems, can help to better understand the locations and severity of the disease before releasing the payload carried by the DDS [94,95,96].

Conde et al. developed AuNPs covered with surface-enhanced Raman spectroscopy (SERS) nano-antennas: 3-3′-diethylthiatricarbocyaniniodid (DTTC) molecules. These SERS reporters were entrapped by modified PEG chains that also served as substrates for cetuximab binding. When evaluated in vivo, the developed theranostic platform presented a high Raman signal, allowing the extensive inhibition of colorectal tumor growth in mice with simultaneous observation using the evaluation of the Raman results [97,98].

Several years later, in 2018, Due et al. reported the development of novel liposomal nanohybrid cerasomes loaded with a novel anti-PD-L1 mAb (atezolizumab), paclitaxel, IRDye800CW and the MRI contrast agent DOTA for dual-modality fluorescence and MRI imaging and CRC treatment. The in vivo experiments performed by these researchers showed that the synthesized cerasomes enhanced tumor contrast and were preferentially accumulated at tumor sites. In addition, they enhanced paclitaxel efficacy, so they may help to detect colorectal tumors more effectively and guide therapeutic decisions [99].

In the same year, in a study also based on liposomal nanohybrid cerasomes, Li et al. described the preparation of a novel EGFR-targeting theranostic nanosystem. Thus, they synthesized porphyrin-containing cerasomes decorated with cetuximab, which were also conjugated with IRDye800CW and the MRI contrast agent DOTA, but with the aim of combining dual-modality tumor detection with photodynamic therapy and immunotherapy. As a result, they found that their theranostic nanosystem provided sensitive detection of tumors in situ and exhibited a high tumor-targeting ability, which suggests that the nanohybrid cerasomes have significant potential for theranostic applications [100].

Finally, Hashemkhani and co-workers investigated the use of cetuximab-conjugated Ag_2_S QDs for the delivery of 5-FU and 5-aminolevulinic acid (ALA) to achieve tumor-specific image-guided photodynamic therapy/chemotherapy combination in EGFR-overexpressing CRC cell lines. To validate the nanoplatform, the authors performed in vitro assays with 2D monolayer cell cultures and 3D spheroids, with which they verified that the nanoplatform had EGFR-targeting capacity, great imaging potential and excellent antitumor properties. In this way, Hashemkhani et al. proposed modifying the nanosystem for targeted imaging and combination therapy for other cancer types in the future [101].

Table 4 provides a summary of all the mAb-loaded nanoplatforms mentioned in this and the previous sections that were developed for CRC diagnosis and/or treatment.

## 8. Nanoplatforms for CRC Therapy in Clinical Trials or Recently Patented

Undoubtedly, nanomedicine is revolutionizing the current development of new cancer diagnosis and therapy systems. This is reflected in the fact that the main drug regulatory agencies are gradually approving the administration of more nanoformulations for the treatment of different types of cancer [41]. However, it is necessary for the scientific community to continue research so that new nanoplatforms enter clinical trials or can be patented, since, for instance, in the specific case of the mAb-loaded nanosystems for CRC diagnosis and therapy, only two have managed to enter a clinical trial or to be patented in recent years (Table 5) [102].

## 9. Conclusions and Suggestions for Future Research

According to the World Health Organization (WHO), 1 million lives were lost to CRC worldwide in 2020. Furthermore, in accordance with the predictions of this organization, the incidence of this disease, which is often diagnosed too late, is expected to continue to increase in the next decades [1,2]. For this reason, there is an urgent need to develop both new strategies that allow CRC detection in the earlier stages, when patients’ chances of survival are greater, and new therapies that are more advantageous than the current ones in terms of specificity.

In this regard, mAb-loaded nanosystems are a valuable tool for the timely detection and targeted treatment of CRC. Thanks to the cellular recognition abilities of mAb, mAb-based nanoplatforms can target tumor cells using two different pathways (passive and active targeting) and contribute to the accumulation of the nanoplatforms in malignant tissues. In addition, these immunoglobulins are not only capable of recognizing molecules expressed on the surface of malignant cells but also various tumor biomarkers. In this sense, it is easy to understand why more and more biosensors, nanoimaging systems, DDS, combined nanotherapies and theranostic nanosystems based on mAb-loaded nanoplatforms are developed each year with the aim of improving the sensitivity of the main CRC diagnostic techniques and reducing the characteristic side toxicity of chemotherapeutics. In the near future, these nanoplatforms will be able to revolutionize CRC nanomedicine. Nonetheless, collective efforts are needed to narrow the existing gap between nanomedicine research and clinical translation, to achieve excellent biocompatibility and great reproducibility and to control the manufacturing process of these nanoplatforms at a large scale.

## Data Availability

Not applicable.

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
