# Peer review of "Antibody-Loaded Nanoplatforms for Colorectal Cancer Diagnosis and Treatment: An Update"

_pharmaceutics, 2023, doi:10.3390/pharmaceutics15051514_

Round 1
Reviewer 1 Report
Review: “Antibody-loaded nanoplatforms for colorectal cancer diagnosis and treatment: An update.” is in general well written and informative. In this article, the authors gave an overview of nanotechnology based new technologies for colorectal cancer early detection. In my opinion, after minor revisions of some typos listed below this paper is acceptable for publication in “Pharmaceutics”.
1. Line 319 Instead of PLC it should be PCL
2. Line 378 Instead IPF it should be IFP
3. Line 508 Instead septin9 it should be septin-9 (same in Table 4)
4. Line 521 Instead anti-claudin7 it should be anti-claudin-7 (same in Table 4)
5. Line 525 Instead ρg/ml it should be ρ (g/mL)
Author Response
RESPONSES TO REVIEWERS’ COMMENTS AND DETAILS OF THE CHANGES THAT HAVE BEEN MADE IN THE MANUSCRIPT: PHARMACEUTICS-2384246
REVIEWER #1
Review: “Antibody-loaded nanoplatforms for colorectal cancer diagnosis and treatment: An update.” is in general well written and informative. In this article, the authors gave an overview of nanotechnology based new technologies for colorectal cancer early detection. In my opinion, after minor revisions of some typos listed below this paper is acceptable for publication in “Pharmaceutics”.
- Line 319 Instead of PLC it should be PCL
- Line 378 Instead IPF it should be IFP
- Line 508 Instead septin9 it should be septin-9 (same in Table 4)
- Line 521 Instead anti-claudin7 it should be anti-claudin-7 (same in Table 4)
- Line 525 Instead ρg/ml it should be ρ (g/mL)
Thank you very much for your comments and for catching the mentioned typos. All of them have already been corrected.

Reviewer 2 Report
Summary / significance: The manuscript by Djermane et al. is a review of the latest literature on the emerging field of nanotechnology in the context of colon cancer (CRC). I congratulate the authors on the effort. The manuscript is well-written and is a good addition to the journal´s readership. The role of using nanomaterials for analysing CRC markers not only on morphological but also on a non-invasive and therapeutic level is becoming increasingly important. This review is informative and easy to read for both the expert and the beginner in the field.
Comment: This article is very extensive, the information of techniques is comprehensive, also showing instructive Figures and Tables, the literature covered is huge, and the level of background is very high.
Specific comments:
1) Some headings and figure legends could be more explicit and detailed. For example, line 364 and line 383, subchapters 4.1. passive targeting and 4.2. active targeting is very short and not very meaningful. Maybe extend to “targeting of tumor tissues”
2) Similarly, the difference between subchapters 5.1. and 5.2. “extracellular” and “circulating” is unclear. Further, in section 5.3 it remains unclear, which imaging techniques are used. This distinction is also not clearly visible in Figure 6, and therefore, the figure could be improved as well by indicating the circulating markers and listing the concrete imaging techniques.
3) the abbreviation DDS (lines 557 and 569), though explained in the introduction, could be written out when mentioned the first time again.

Essentially, the English phrasings needs thorough improvements. often the word order is inappropriate and some words are very unusual (apparition, epithermal, popularly, interstice, …). I have uploaded an edited pdf version with a number of proposed changes. If possible, the article should be edited and proofread by an English native speaker or expert.
Author Response
RESPONSES TO REVIEWERS’ COMMENTS AND DETAILS OF THE CHANGES THAT HAVE BEEN MADE IN THE MANUSCRIPT: PHARMACEUTICS-2384246
REVIEWER #2
Summary / significance: The manuscript by Djermane et al. is a review of the latest literature on the emerging field of nanotechnology in the context of colon cancer (CRC). I congratulate the authors on the effort. The manuscript is well-written and is a good addition to the journal´s readership. The role of using nanomaterials for analysing CRC markers not only on morphological but also on a non-invasive and therapeutic level is becoming increasingly important. This review is informative and easy to read for both the expert and the beginner in the field.
Comment: This article is very extensive, the information of techniques is comprehensive, also showing instructive Figures and Tables, the literature covered is huge, and the level of background is very high.
Thank you very much for taking the time to review our work and for all your comments and suggestions. We really appreciate them. All changes we have made in the manuscript have been marked in red.
Specific comments:
1) Some headings and figure legends could be more explicit and detailed. For example, line 364 and line 383, subchapters 4.1. passive targeting and 4.2. active targeting is very short and not very meaningful. Maybe extend to “targeting of tumor tissues”.
According to your suggestion, the heading of the 4.1 and 4.2 points has been extended, as well as the legends of figures 1 and 6.
2) Similarly, the difference between subchapters 5.1. and 5.2. “extracellular” and “circulating” is unclear. Further, in section 5.3 it remains unclear, which imaging techniques are used. This distinction is also not clearly visible in Figure 6, and therefore, the figure could be improved as well by indicating the circulating markers and listing the concrete imaging techniques.
Based on your comment, the heading of point 5.2. has been modified so that extracellular and circulating biomarkers can be easier differentiated. Moreover, information about the imaging techniques employed in the research works mentioned in section 5.3. has been included, and figure 6 has been completed.
3) The abbreviation “DDS” (lines 557 and 569), though explained in the introduction, could be written out when mentioned the first time again.
Following your recommendation, in addition to explaining DDSs meaning in the introduction (line 121), we have indicated it again in line 286, when it appears for the second time.
Comments on the Quality of English Language
Essentially, the English phrasings needs thorough improvements. often the word order is inappropriate, and some words are very unusual (apparition, epithermal, popularly, interstice, …). I have uploaded an edited pdf version with a number of proposed changes. If possible, the article should be edited and proofread by an English native speaker or expert.
Thank you very much for all the proposed changes, all of them have already been made. In addition, the manuscript has been sent to the MDPI English editing service to be checked.

Reviewer 3 Report
The manuscript entitled “Antibody-loaded nanoplatforms for colorectal cancer diagnosis 2 and treatment: An update”. It is a very well designed and written manuscript about the nanoplataforms with antibody loaded drugs in diagnosis and treatment. The authors provided a state of the art of the colorectal cancer and focused in the new knowledge on nanosystems. I believe the manuscript can be accepted in the current form.
Author Response
RESPONSES TO REVIEWERS’ COMMENTS AND DETAILS OF THE CHANGES THAT HAVE BEEN MADE IN THE MANUSCRIPT: PHARMACEUTICS-2384246
REVIEWER #3
The manuscript entitled “Antibody-loaded nanoplatforms for colorectal cancer diagnosis and treatment: An update” is a very well designed and written manuscript about the nanoplataforms with loaded antibodies and drugs used in diagnosis and treatment. The authors provided a state of the art of colorectal cancer and focused in the new knowledge on nanosystems. I believe that the manuscript can be accepted in the current form.
Thank you very much. We really appreciate your comments.
